# Electronic reminders and rewards to improve adherence to inhaled asthma treatment in adolescents: a non-randomised feasibility study in tertiary care

Anna De Simoni ![ORCID],[1] Louise Fleming,[2] Lois Holliday,[1] Robert Horne,[3] Stefan Priebe,[4] Andrew Bush,[2] Aziz Sheikh,[5] Chris Griffiths ![ORCID] [1]

For numbered affiliations see end of article.

**Correspondence to**
Dr Anna De Simoni;
a.desimoni@qmul.ac.uk

## ABSTRACT

**Objective** To test the feasibility and acceptability of a short-term reminder and incentives intervention in adolescents with low adherence to asthma medications.

**Methods** Mixed-methods feasibility study in a tertiary care clinic. Adolescents recruited to a 24-week programme with three 8-weekly visits, receiving electronic reminders to prompt inhaled corticosteroid (ICS) inhalation through a mobile app coupled with electronic monitoring devices (EMD). From the second visit, monetary incentives based on adherence of ICS inhalation: £1 per dose, maximum £2/day, up to £112/study, collected as gift cards at the third visit. End of study interviews and questionnaires assessing perceptions of asthma and ICS, analysed using the Perceptions and Practicalities Framework.

**Participants** Adolescents (11–18 years) with documented low ICS adherence (<80% by EMD), and poor asthma control at the first clinic visit.

**Results** 10 out of 12 adolescents approached were recruited (7 males, 3 females, 12–16 years). Eight participants provided adherence measures up to the fourth visits and received rewards. Mean study duration was 281 days, with 7/10 participants unable to attend their fourth visit due to COVID-19 lockdown. Only 3/10 participants managed to pair the app/EMD up to the fourth visit, which was associated with improved ICS adherence (from 0.51, SD 0.07 to 0.86, SD 0.05). Adherence did not change in adolescents unable to pair the app/EMD. The intervention was acceptable to participants and parents/guardians. Exit interviews showed that participants welcomed reminders and incentives, though expressed frustration with app/EMD technological difficulties. Participants stated the intervention helped through reminding ICS doses, promoting self-monitoring and increasing motivation to take inhalers.

**Conclusions** An intervention using electronic reminders and incentives through an app coupled with an EMD was feasible and acceptable to adolescents with asthma. A pilot randomised controlled trial is warranted to better estimate the effect size on adherence, with improved technical support for the EMD.

## Strengths and limitations of this study

► UK study on adolescents diagnosed with asthma of different ethnic backgrounds, exploring feasibility and acceptability of electronic reminders through an electronic monitoring devices (EMD) coupled with a mobile app system and financial incentives to improve adherence to inhaled preventer medications.
► Objective adherence documentation through EMD monitoring.
► Intervention developed with patient and public involvement and informed by the Perceptions and Practicalities Framework framework.
► A limitation was the small number of participants, technical problems with the EMD-app system, and no long-term asthma outcome data.

## BACKGROUND

Asthma is the most common chronic disease of adolescence in the UK.[1] Asthma prevalence, morbidity and mortality are high among adolescents, especially in urban, ethnic minority adolescents.[2 3] Reported adherence to inhaled corticosteroid (ICS) in adolescents is poor, ranging from 25% to 35%,[4 5] and results in adverse outcomes, including death.[6 7] Greater than 80% adherence to ICS has been shown to reduce asthma exacerbations.[8] New approaches to improve adherence to ICS treatment in adolescents are urgently needed.[9]

Electronic reminder systems have the potential to improve adherence; however, a systematic review has shown only a weak-to-moderate impact,[10] with effects waning after 4 weeks, which likely reflect habituation to reminders.[11] A growing body of evidence suggests financial incentives can augment adherence; public attitudes to incentives are largely positive, especially when targeted

at groups with particular health needs.[12–14] A systematic review and meta-analysis of studies of incentives showed positive health benefits.[15] Importantly, short-term (3 months) financial incentives have been linked with sustained (1 year) health behavioural change in adolescents with diabetes, suggesting that incentives can establish beneficial behaviour patterns which persist even when the incentive is withdrawn.[16] Our group completed the first trial in the UK testing financial incentives, showing improved adherence to medication for people with major mental health problems.[17] Incentives can potentially improve asthma medication adherence—in a recent pilot study, electronic reminders coupled with incentives improved daily adherence to ICS inhalers in African American adolescents.[18]

According to Horne's Perceptions and Practicalities Approach (PAPA)[19] interventions ought to address both perceptions and practicalities that influence motivation and ability to adhere, through conscious and unconscious processes. A period of sustained adherence to ICS dosing could lead to improved asthma control which the adolescent may associate with regular ICS use, thereby altering attitudes and behaviour regarding long-term adherence to ICS.

We, therefore, set out to explore the feasibility of a reminders and incentives intervention to improve asthma inhaler adherence in adolescents attending a severe asthma clinic in tertiary care. The research team had already explored adherence in adolescents using Hailie sensor (Adherium, New Zealand)[20] electronic monitoring devices (EMD), distinguishing adolescents who have therapy resistant asthma from the ones who are poorly adherent.[21] Participants from the latter group were recruited to the study.

Our overarching hypothesis was that a short-term (8 weeks) reminder and incentives intervention has the potential to deliver improvement in adherence and in clinical end-points in adolescents.

The questions we planned to answer in this initial feasibility study were:
1. Is using reminders and incentives to promote adherence acceptable to adolescents and parents/guardians?
2. Can we recruit adolescents with poorly controlled asthma with documented poor adherence to ICS inhaler treatment?
3. Is the process of monitoring and rewarding adolescents technically feasible?
4. What level of financial incentive appears optimal in practice (as opposed to consultation with patient and public involvement (PPI))?
5. Can we successfully deliver rewards to adolescents?

## METHODS

This was a mixed-methods feasibility study in a tertiary care outpatient clinic. Participants enrolled in a 24 weeks non-randomised feasibility study with three 8-weekly visits. The study ran between September 2018 and May 2020.

## Participants

Eligible patients attending the Royal Brompton Hospital were approached by a clinician who was part of the direct care team, asking for patients' and parental consent to be contacted by a researcher. Once patients consented, they were contacted by the study team and fully informed about the study. Age-appropriate assent from adolescents and written consent to take part in the study were obtained from parents/guardian.

### Inclusion criteria

Inclusion criteria were:
► Adolescents with asthma diagnosed on standard criteria[22] attending outpatient asthma clinics in hospital.
► Aged between 11 and 18 years.
► Poor adherence (<80%) measured in the previous year over a period of at least 3 months, using an EMD.
► Own/use of a smartphone.
► Evidence of poor asthma outcomes at presentation at the clinic, measured as Asthma Control Test (ACT) score <20, or ≥2 asthma attacks in the previous year or high fractional exhaled nitric oxide (FeNO) at presentation at the clinic at the first visit.

### Exclusion criteria

► Patients considered by hospital clinician to be unsuitable for the project for any reason.
► Not accompanied by parents if <18 years.

## Intervention

### Patient and public involvement

PPI was collected through a workshop with children and young people (n=8) part of the Asthma UK Centre for Applied Research PPI group and an online survey (SurveyMonkey).[23] The link to the online survey was shared on the Asthma UK Facebook page (88 PPI with asthma: adolescents, 28 parents of adolescents with asthma, 51 adults).

All adolescents agreed the intervention could be helpful in improving adherence to inhaler treatment, with the great majority indicating £1/dose as appropriate and gift cards as way to collect rewards. They felt the incentives would increase motivation for taking inhalers and partly work as reminder.

Adolescents who took part in the workshop welcomed the idea of research to explore the effects of electronic reminders and rewards to improve treatment adherence. The potential benefits they saw—specifically for adolescents—included increased motivation to take medication, help with remembering to use inhalers and development of a good routine.

Similarly, all adolescents in the survey agreed the intervention could be helpful in improving inhaler treatment, with the great majority indicating £1/dose as appropriate and gift cards as way to collect rewards. They felt the incentives would increase motivation for taking inhalers and partly work as reminder. Their concerns focused around

**Table 1** Study visits, intervention and outcomes

| Time point | Study visits | | | |
| | Visit 1 week 0 | Visit 2 week 8 | Visit 3 week 16 | Visit 4 week 24 |
| --- | --- | --- | --- | --- |
| **Enrolment** | | | | |
| Informed consent | X | | | |
| **Intervention** | | | | |
| Financial incentives | | X | | |
| Electronic reminders and monitoring via Smartinhalers | X | X | X | X |
| **Assessments** | | | | |
| Primary outcome: % of patients enrolled, % patients with successfully adherence data, % of patients who collected financial rewards | X | X | X | X |
| Secondary outcomes: adherence to ICS, ACT score, illness perception questionnaire, FeNO, FEV1 | X | X | X | X |
| BMQ B-IPQ | X | X | X | X |
| Semistructured Interviews (patients) | | | | X |

ACT, Asthma Control Test; B-IPQ, Brief Illness Perception Questionnaire; BMQ, Beliefs about Medicines Questionnaire; FeNO, fractional exhaled nitric oxide; FEV1, forced expired volume; ICS, inhaled corticosteroid.

the need to prove that teenagers actually inhaled the medication, the linked requirement of funding for the incentives; queries about the funder for the incentives; and whether the reward could lead to inhaler overuse.

Parents' feedback was similar to teenagers, with the additional concern regarding a diminished sense of responsibility for management of asthma, teenagers potentially stopping taking inhalers once the intervention ended and practical barriers such as mobile phones being switched off (eg, at school).

## Intervention

Our 24-week intervention consisted of reminders for ICS inhalation through a mobile app and 8 weeks incentives proportional to adherence between visits 2 and 3 (see table 1). The intervention was delivered through an EMD-app system as an adjunct to usual care. The intervention was based on the PAPA framework and informed by evidence from a previous qualitative study,[24] stakeholders' consultations and PPI.

At the first study visit, participants were issued with EMD (Smartinhalers) for their ICS inhaler. Under the researcher's guidance, adolescents downloaded the Hailie app (Hailie app and sensor (Adherium, New Zealand),[17] paired their EMD with the app and received face-to-face training on the EMD-app system. Medication use data were visible in the Hailie App via data transfer (via Bluetooth) and uploaded to the Hailie Web Portal via a secure internet connection (cellular data/Wi-Fi). In the online portal, patients' details were replaced with a number code.

Daily audio reminders were programmed by adolescents themselves on the Hailie app. Reminders could be set to come from the EMD or the app, at patient's choice. The 8-week period when incentives were offered started from the second visit, during which patients were informed about the start and end date of their incentive phase. Incentive payments were given as Google Play gift cards, Apple-iTunes or Amazon gift cards, according to participant choice and the adherence data downloaded via the EMD system. Participants were informed about the importance of taking their inhaler correctly in respect of number of doses and prescribed time (am and/or pm) for adherence to be calculated correctly (ie, taking the prescribed two puffs am and two puffs pm as a single dose of four puffs once daily would have resulted in adherence of 50%). The incentive amount was capped at £2/day; taking the inhaler more than prescribed would not therefore have increased the incentive gained. The maximum amount of the gift cards was £112 (£2/day × 56 days, ie, 8 weeks). Participants were informed about the calculation. Gift cards were handed over to participants at the third visit. Adolescents aged 13 years and older were able to redeem Google and Apple Store gift cards/vouchers, while Amazon vouchers needed be redeemed through a parent/guardian account.

When the paring EMD-Hailie app failed between study visits, adherence data were downloaded directly from the EMD via Hailie Connect using a USB (Universal Serial Bus) connection at the subsequent clinic. Technical support for the EMD-Hailie app system was available via email to research staff. There was no customer/patient support, apart from the 'Help' section within the app itself.

After collecting the reward at the third visit, participants received the reminders intervention for a further 8 weeks up to visit 4, to verify sustainability of the effects of incentives on study outcomes.

At each visit, participants completed an ACT score, FeNO, spirometry (first second forced expired volume), demographics, asthma history, Brief Illness Perception Questionnaire (B-IPQ),[25] Beliefs about Medicines Questionnaire (BMQ).[26] At the final visit, participants took part in a semistructured exit interview.

## Study analysis and outcomes

This was a feasibility study, which did not require a formal sample size calculation. The target number of participants (10) was based on a resource constraint, that is, the number of adolescents followed up at the clinic (around 30) with prior adherence measured using an EMD.

**Table 2** Participants' characteristics

| N. | Gender | Age | Atopy Y/N | EMD adherence before the study, % | EMD adherence during reminders+incentives, % | Exit interview Y /N |
|---|---|---|---|---|---|---|
| 1 | F | 11–14 | Y (eczema, food allergy, hay fever) | 46 | 44 | Y |
| 2 | M | 15–18 | Y (eczema, hay fever) | 74 | 77 | Y |
| 3 | M | 11–14 | Y (eczema, food allergy, hay fever, history of anaphylaxis) | 59 | 68 | Y |
| 4 | M | 11–14 | Y (food allergy) | 70 | n/a | N |
| 5 | F | 11–14 | Y (hay fever) | 3 | n/a | N |
| 6 | M | 15–18 | Y (food allergy, hay fever) | 49 | 83 | Y |
| 7 | M | 15–18 | – | 45 | 82 | N |
| 8 | M | 11–14 | Y (eczema, hay fever) | 59 | 92 | Y |
| 9 | F | 11–14 | Y (food allergy, anaphylaxis, eczema, hay fever) | 54 | 58 | N |
| 10 | M | 15–18 | Y (food allergy, anaphylaxis, eczema, hay fever) | 72 | 58 | Y |

EMD, electronic monitoring device; F, female; M, male; N, no; n/a, not applicable; Y, yes.

The following feasibility and acceptability criteria were chosen to determine whether this work should proceed to a pilot randomised controlled trial (RCT):

1. Acceptability of the intervention to adolescents and parents/guardians.
2. Recruitment of patients approached and retention in the study ≥75%.
3. Successful data gathering.
4. Evidence from applying PAPA framework model that this approach has the potential to deliver improvements in attitudes to long term medication adherence.

We used GraphPad Prism V.9.0.1 (GraphPad, La Jolla, California, USA) for statistical analysis. Adherence data were evaluated using one sample t-test. A $p<0.05$ was regarded as statistically significant.

Secondary outcomes included: ACT score, BMQ, B-IPQ FeNO, spirometry, demographics and asthma history. Due to the limited number of data collected and the lack of power to detect trend and significance, no statistical analysis was performed on these outcomes.

Interviews were analysed using thematic analysis as described by Braun and Clarke,[27] using the PAPA framework.[20]

## RESULTS

Patients' mean age was 14.2 years (SD 1.3, age range 12–16 years), 6 participants were <15 years old, 7/10 male, from a variety of ethnic groups: 2 blacks, 1 black Caribbean, 3 Caucasians, 2 Asians, 1 mixed Caucasian/Asian, 1 mixed Caucasian/black (table 2).

### Participation and retention in the study

Twelve patients were identified as eligible and invited to participate in the study. Of those invited, 10 patients (80% participation response rate) were recruited to the study, 1 parent declined on behalf of the child due to lack of time and having to rush off the clinic, 1 patient did not own a personal mobile phone.

Of the 10 participants, 1 patient attended only the first visit and was lost to follow-up (they were not brought to subsequent clinic visits). Another participant was unable to pair the device and app between visits 1 and 2, subsequently failed twice to bring the EMD to the clinic and withdrew from the study.

Eight participants provided adherence measures up to the fourth visit and received rewards (table 3). Patients provided complete and valid outcome data secondary outcomes up to the third visit. Due to clinics switching to

**Table 3** Summery outcomes collected during the study

| | Adherence | ACT | FeNO | FEV1 |
|---|---|---|---|---|
| Visit 1 | | 18.50 (5.23) n=10 | 44.25 (21.92) n=8 | 2.39 (0.50) n=10 |
| Visit 2 | 0.66 (0.18) n=9 | 19.00 (6.28) n=5 | 33.42 (28.93) n=7 | 2.65 (0.67) n=7 |
| Visit 3 | 0.64 (0.28) n=8 | 20.75 (2.06) n=4 | 45.40 (37.15) n=5 | 2.69 (0.85) n=6 |
| Visit 4 | 0.52 (0.24) n=8 | 19.50 (0.70) n=2 | 67.00 (15.55) n=2 | 1.92 n=1 |

ACT, Asthma Control Score; FeNO, fractional exhaled nitric oxide; FEV1, forced expiratory volume.

remote during the COVID-19 pandemic, five patients did not provide outcome data beyond the third visit.

The mean study duration was 40±10 (SD) weeks, compared with the 24weeks set in the protocol (see table 1). The longer duration was due to three main factors. First, when participants encountered technical difficulties with pairing the EMD-app (mostly between the first and the second visit), the incentive phase was postponed to the subsequent clinic visit. The issue was either fixed or a new EMD provided to patients. If pairing could not be achieved, participants were informed that adherence data would be downloaded at the subsequent clinic via USB. Second, logistic difficulties were experienced by both the clinic-booking staff and participants' parents/guardians with 8weeks follow-up clinics. The average interval between clinics was about 3months. Third, some participants were unable to attend some of the appointments, which were then re-booked at later dates, therefore lengthening the total duration of participation in the study.

### Delivery of incentives

All participants chose to receive incentives as Amazon gift cards, which were handed out at the third visit or sent in the post. Participants received a median of £82 (range £50–£104). No patients expressed concerns about collecting their incentives via their parents' Amazon account, about the amount awarded or any other aspects. At the fourth visit and exit interviews, patients showed appreciation of the gift cards.

### OUTCOMES

Throughout the study, overall adherence to ICS did not significantly change (table 3, figure 1, L bottom panel). At monitoring prior to entering the study, mean adherence was 0.53 (SD 0.20, n=10), 0.66 at visit 2 (SD 0.17, n=9), 0.64 at visit 3 (SD 0.27, n=8) and 0.51 at visit 4 (SD 0.24, n=8). For the three participants able to pair the EMD with the app, receive reminders and self-monitor via the app throughout the study, adherence with reminders and incentives was higher at 0.86 (SD 0.05) compared with baseline 0.51 (SD 0.07) (table 3, figure 1, R bottom panel).

The timing of collection of outcomes was different for adherence (collected remotely via the online portal or through a USB cable at the clinic) compared with the other outcomes that had to be collected at the clinic visit (ACT score, BMQ, B-IPQ, FeNO, spirometry, demographics, asthma history). This was associated with a variable lag of time between the end of the incentive period

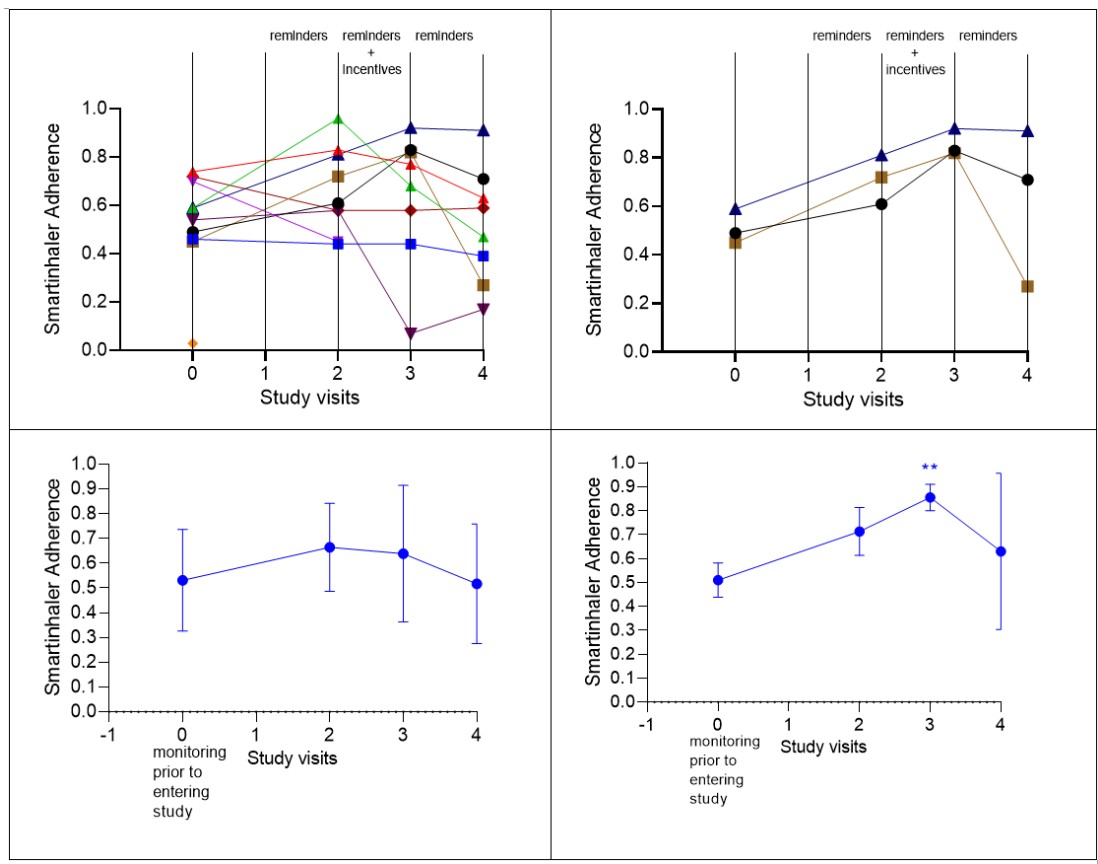

**Figure 1** Participants' adherence at study visits (2, 3 and 4) compared with adherence measured duringmonitoring prior to entering the study (time 0). L: individual adherence data (top panel); pooled mean and SD (bottom panel). R: individual adherence data for the three participants able to pair app-smartinhaler and self-monitor ICS taking (top panel); pooled mean and SD (bottom panel). ICS, inhaled corticosteroid.

**Table 4** Beliefs about Medicines Questionnaire (BMQ)

| Belief statements | Baseline median score (n=10) | Baseline agreeing or strongly agreeing (%) | Visit 3 median score Visit 3 (n=7) | Visit 3 agreeing or strongly agreeing (%) |
|---|---|---|---|---|
| (1) My health depends on asthma medicines | 2 | 60 | 3 | 29 |
| (2) Having to take asthma medication worries me | 4 | 0 | 4 | 0 |
| (3) My life would be impossible without my asthma medication | 2 | 60 | 3 | 29 |
| (4) Without my asthma medication I would be very ill | 2 | 70 | 3 | 57 |
| (5) I sometimes worry about the long-term effects of my asthma medication | 3.5 | 40 | 3 | 14 |
| (6) My asthma medication is mystery to me | 4 | 10 | 4 | 14 |
| (7) My health in the future will depend on my asthma medication | 3 | 20 | 3 | 14 |
| (8) My asthma medication disrupts my life | 4 | 0 | 4 | 14 |
| (9) I sometimes worry about becoming too dependent on my asthma medication | 3 | 40 | 3 | 14 |
| (10) My asthma medication protects me from becoming worse. | 2 | 90 | 2 | 71 |
| (11) Doctors use too many medicines | 4 | 30 | 3 | 29 |
| (12) People who take medicines should stop their treatment for a while every now and again | 3 | 30 | 3 | 43 |
| (13) Most medicines are addictive | 4 | 10 | 3 | 0 |
| (14) Natural remedies are safer than medicines | 3 | 30 | 3 | 43 |
| (15) Medicines do more harm than good | 4 | 10 | 3 | 14 |
| (16) All medicines are poisons | 5 | 10 | 4 | 0 |
| (17) Doctors place too much trust on medicines | 4 | 20 | 3 | 29 |
| (18) If doctors had more time with patients, they would prescribe fewer medicines | 3 | 10 | 3 | 14 |

Likert responses for the BMQ were scored 1 (strongly agree) to 5 (strongly disagree). A lower score represents more agreement with the item. Shaded columns refer to visit 3, that is, on receiving incentives.

(which was exactly 8 weeks since visit 2) and the subsequent collection of secondary outcomes at the clinic.

**Insight from PAPA framework: BMQ, B-IPQ and exit interviews**
Scores of individual items from the BMQ-concerns scale indicated that 40% of patients were worried about the long-term effects of their asthma medication and 40% were concerned about becoming too dependent on them (table 4). The percentages for both these items decreased to 14% after the incentives.

According to the BMQ, >60% of patients at baseline agreed or strongly agreed about the necessity of taking asthma medications, while this somewhat decreased after incentives to 30% (see complete row data in online supplemental table 1).

Illness perceptions scores from the B-IPQ are shown in table 5. This is in keeping with the trend of increased control on asthma, decreased concerns, decreased emotional impact and effects on life emerging from the B-IPQ.

Of 10 patients, 6 had exit interviews, of which two were face to face, while four were by telephone due to COVID-19 lockdown. Most participants felt the intervention helped with increasing regularity of taking ICS inhalers and improving overall management of asthma (table 6).

This was mainly down to facilitation with practical aspects of taking ICS, such as reminding doses and ability of self-monitoring through the app (table 7). Technical issues with the EMD-app were reported as the cause of no such effect for one participant (N.3).

Technical problems included: inability to pair EMD with the app; reminders coming from the EMD despite being set to be delivered by the app; no reminders audible despite being properly setup, the length of time taken for the paring EMD-app via Bluetooth. Interestingly, technical difficulties were reported as practical issues hindering participants' capability of taking ICS.

 De Simoni A, *et al. BMJ Open* 2021;11:e053268. doi:10.1136/bmjopen-2021-053268

**Table 5** Brief Illness Perception Questionnaire

| Participant | Baseline mean score (n=10) | Baseline SD (n=10) | Visit 3 mean score (n=7) | Visit 3 SD (n=7) |
|---|---|---|---|---|
| How much affects your life | 4.6 | 2.5 | 3.7 | 2.3 |
| How long your illness will continue (0=short time, 10=forever) | 6.1 | 2.4 | 4.6 | 1.4 |
| How much control do you have | 6.2 | 2.5 | 6.6 | 2.2 |
| How much treatment can help | 7.7 | 1.9 | 7.0 | 2.6 |
| How much symptoms you experience | 5.1 | 1.9 | 4.6 | 2.2 |
| How concern you are | 5.2 | 2.6 | 3.6 | 2.6 |
| How well you understand your illness | 8 | 1.8 | 8.4 | 1.8 |
| How much affects you emotionally | 2.7 | 2.1 | 1.6 | 1.8 |

0=not at all, 10=extremely. A higher score represents more agreement with the item.
Shaded columns refer to visit 3, that is, on receiving incentives.

In terms of perceptions, incentives were specifically associated with increased motivation of taking ICS (table 7). This was in keeping with the trend of increased control on asthma, decreased concerns, decreased emotional impact and effects on life emerging from the B-IPQ (table 5).

All participants felt 8 weeks and £1/ICS dose were an appropriate length of time for the incentive phase and stated they would have taken part in a study with a randomisation process to decide whether or not they were receiving the intervention.

None of the interviewees felt that the intervention had changed their views on the importance of taking ICS, despite a trend in increased treatment control within the B-IPQ (table 5).

## DISCUSSION
### Summary
Participation and retention in the study were both 80%. The reminders and incentive intervention were acceptable to both adolescents and their parents/guardians. Rewards of £80/participant on average were successfully delivered. Data collection was feasible despite the COVID-19 pandemic cutting the collection of secondary outcomes to the third visit for most participants. Findings from the BMQ, B-IPQ and exit interviews suggest reminders, incentives and the ability of self-monitoring ICS were practical facilitators of ICS taking. In respect of patients' perceptions, interview data suggest incentives were associated with increased motivation for adolescents to take inhalers. Thus, the intervention has the potential to deliver improvements in ICS adherence.

The main obstacle encountered in the study were the technical difficulties with the EMD-app system, which limited the assessment of the intervention effect on outcomes. Participants expressed frustration at the limitations of technology and stated the intervention would have helped to a greater extent if the EMD-app system worked correctly throughout the study. Despite these limitations, results point to potential positive effects on improved ICS adherence measured through the EMD, though more data are needed to estimate the effect size and sustainability of the effect after stopping incentives.

### Strengths and limitations
A strength of the study is the availability of previous adherence through EMD data, which enabled recruitment of adolescents whose poor control was likely due to poor adherence. The intervention was developed through different PPI activities, is based on our previous findings on factors affecting adherence to asthma inhalers in adolescents[20] and informed by the PAPA framework. The setting of the study was a real world, ethnically diverse and busy asthma clinic, strengthening findings on feasibility of the intervention and collection of outcomes.

The small number of participants, the lack of measurements of the intervention effects on asthma in the long term are important limitations. Technical problems with the EMD-Hailie app system and the lack of customer support limited the assessment of the effect of the intervention on all study outcomes. The lack of reliability of reminders delivery within some EMDs may have had significant effects on the study outcomes, calling for further research with more reliable EMDs prior to a definitive RCT. The time lag between collection of secondary outcomes (next clinic visit) in relation to EMD adherence (which was collected through the online portal, with patients aware of when their incentive period end date) could have introduced bias, especially in respect to outcomes at the third visit. The retention rate to the study (80%) may suggest that that even a cash incentive is not enough for some teenagers to engage/take inhalers.

COVID-19 lockdown prevented collecting secondary outcomes for the last study visit. Interviews scripts were relatively short and limited in data richness. This could be down to face-to-face interviews being undertaken in a quiet corner of an otherwise busy outpatient clinic and the intrinsic difficulties of performing 1:1 telephone interviews with adolescents, some time since the last contact at the clinic (due to the COVID-19 lockdown and the wait

**Table 6** Exit semistructured interviews

| N. | Satisfaction | Help with increasing regularity taking ICS inhalers | Help with improving overall management of asthma | App-smartinhaler connection problems Y/N | Most useful part | Least useful part | How long rewards to be offered for maximum help | Did it alter views about the importance of your medicines Y/N | Would you take part in this study if randomisation to intervention Y/N |
|---|---|---|---|---|---|---|---|---|---|
| 1 | Moderately satisfied | Quite a bit | Quite a bit | Y | Both reminders and incentives | Reminders at times annoying, switched them off at times | 8 weeks is enough time | N | Y |
| 2 | Neither satisfied nor dissatisfied | Quite a bit | Very little | Y | Both reminders and incentives | / | 8 weeks is enough time. | N | Y |
| 3 | Moderately satisfied | Not at all | Very little | Y | Tracking adherence through the app | / | 8 weeks is the right amount of time | N | Y |
| 4 | Declined | | | Y | | | | | |
| 5 | Lost to f/u | | | | | | | | |
| 6 | Very satisfied | Quite a bit | Very little | N | Both reminders and incentives | n/a | 8 weeks is the right amount of time | N | Y |
| 7 | Not contactable | | | N | | | | | |
| 8 | Very satisfied | Extremely | Extremely | N | The Incentives | / | 8 weeks is the right amount of time | n/a | Y |
| 9 | Not contactable | | | Y | | | | | |
| 10 | Moderately satisfied | Quite a bit | Somewhat | Y | Both reminders and incentives. Tracking adherence through the app. | / | 8 weeks is the right amount of time | n/a | Y |

All questions refer to the reminders and incentive intervention.
f/u, follow-up; ICS, inhaled corticosteroid; N, no; n/a, not applicable; Y, yes.

**Table 7** Thematic analysis of exit interviews analysed according to the PAPA framework

| Barriers | Facilitators |
|---|---|
| **Practicalities**<br>Treatment capability and resources | **Practicalities**<br>Treatment capability and resources |
| Technical problem with the EMD system:<br>► Issues with pairing app-EMD<br>*'Technical issues with the app not pairing with the device.*<br>*Pairing with the phone and the device was quite important. A lot of the time it didn't pair at all and data was lost'. N.10*<br>► Reminders from EMD, not the app<br>*'Despite correct alarm set-up, the reminders only came from the device. It woke me up many times during the night, the flashing and the noise, doesn't really help'. N.3*<br>► No reminders<br>*'Technical issues with app never sending reminders. To maximise incentives, I set an alarm on my phone as reminder'. N.8*<br>► Time taken to load EMD data to app<br>*'After a while the app started taking ages to load inhaler use records. I just stopped doing it because of how long it was taking'. N.2* | ► Reminders help with remembering taking ICS<br>*'It helped because it was an actual reminder to actually take my medicine' N.2*<br>*'It didn't really change, I've always known that it's important, it just was maybe a bit more encouraging to take it or maybe not encouraging but it just reminded me to take it more.' N.2*<br>► Feedback on inhaler taking through self-monitoring<br>*'I could look on my phone, it would be there which was useful to see, how many times I was taking it'. N.3*<br>*'They were incredibly helpful. It helped keeping track of time'. N.6*<br>*'[What would help ICS adherence] Being able to keep tracking my inhaler use after the end of the intervention. If the app did connect, I would still remember to take it every day'. N.8*<br>*[What would help ICS adherence] 'The recording was quite good because you can see yourself if you're doing it correctly, yeah, I think that was a good method of making sure that you are still taking it'. N.10*<br>► Suggestions for improvement<br>*Fixing technical issues with the app [would have helped]. An app that's convenient to check and see how you're doing and it was quick to pair and everything, just a smooth firing app would just be good, especially for younger people' N.2*<br>*'Adding the vibration function for the reminders from the app [would improve ICS adherence]'. N.6* |
| **Perceptions**<br>Treatment necessity and concerns | **Perceptions**<br>Treatment necessity and concerns |
| Necessity beliefs<br>► No change<br>*'It is still the same'[did not alter views about asthma and treatment] N.1*<br>*'No, (did not alter views about asthma and treatment) but I already knew it was important to take it anyway so'. N.3*<br>*'Quite a little because prior to this my asthma was already very in control'(did not alter views about asthma and treatment). N.6*<br>Concerns n/a | Necessity beliefs<br>► Being monitored<br>*'Knowing that someone was going to check up made me take ICS more, and more conscious about ICS'*<br>Concerns<br>► Incentives increase motivation<br>*'I think that was a generous amount. It definitely, the incentive part definitely was the most consistent I've ever been [with ICS]. Had it continued I probably would have been more consistent but it's a long time to give money to people for just taking medicine, so I think eight weeks is good.' N.2*<br>*'I think the eight weeks are just fine, it gives enough time to actually get used to it. And the reward is incentivising enough'. N.6*<br>*"Being very selfish, [what helped the most was] being paid. I'm already very regular on my inhaler so it wasn't much of a difference, the reminders were very helpful because I could take it on time". N.6*<br>*'[Most useful part] 'I think it was equally, to notify me and remind me to do my inhaler but I think a younger age group would probably prefer getting actually paid because they could get actually something with that money and hopefully it would actually encourage them to not forget and take the medication'. N.10* |

EMD, electronic monitoring device; ICS, inhaled corticosteroid; PAPA, Perceptions and Practicalities Framework.

for approval from the Research Ethics Committee of the amendment to switch to phone interviews).

### Comparison with existing literature

These data are consistent with previous evidence about the importance of reminders for ICS adherence,[11] and the positive effects of incentives on medication adherence.[14] None of the concerns raised by adolescents and parents in our PPI activities (see the PPI section in the Methods section) or in the wider literature[28] came up in this feasibility study when talking to adolescents and their parents or in the exit interviews.

EMDs hold potential to support adolescents' adherence,[14 29 30] and can therefore be used to empower adolescents and enable them to monitor their adherence and promote self-management. Electronic dose reminding and monitoring effects on adherence might not be sustained over time,[14] and therefore, there is a need to find new ways to sustain such effects. The incentive

component was added in the intervention in the attempt to increase motivation of taking ICS inhalers and sustain the intervention effect over time. Although only data of few participants were available, interviews data suggest incentives were associated with increased motivation for adolescents to take inhalers. Adolescents stated the intervention helped with practical aspects of taking medications through getting reminders and the ability of self-monitoring, which were indeed identified as facilitators to adherence in a previous study of UK adolescents.[20]

### Clinical and research implications

While the effects of reminders on ICS adherence have been previously documented, the feasibility, acceptability and relatively easy delivery of financial incentives in teenagers warrant clinical and research attention.

The reported effects on motivation of taking ICS inhalers is promising and needs further investigations.

A previous study showed complex relations between illness perceptions, medication beliefs, medication adherence, disease control and quality of life in adolescents with asthma.[31]

A pilot RCT is warranted to estimate the effect size on adherence, with improved technical support for the EMD. Future focus groups with study participants could be used to further collect in-depth data regarding the acceptability of financial incentives, perceptions of the intervention, whether the incentives could be applied in a different way (eg, rewards for reaching targets) and whether there are other additional factors that could be built into the intervention to motivate better adherence (eg, a visualisation of rewards to date; push notifications to the participants social media account). Parents and healthcare professionals' views also need to be captured.

Remote questionnaire filling and collection of study outcomes that do not require clinic attendance (such as ACT, BMQ and B-IPQ) could improve timing of data collection by narrowing the lag between adherence and other study outcomes.

### CONCLUSIONS

This study showed that using reminders and incentives to promote adherence is acceptable to adolescents and parents/guardians. Adolescents with poorly controlled asthma and documented poor adherence to ICS inhaler treatment can be recruited from a tertiary care clinic. Adherence can be monitored using cloud technology and outcome data collected on adherence and asthma control, though would benefit of an improved EMD-app system and customer support service. £1/ICS dose for 8 weeks appears promising in practice and is easily delivered to adolescents. A pilot RCT is warranted to better estimate the effect size on adherence. Adolescents with asthma recruited in the study engaged with exit interviews, valued participation in the study and would consider randomisation in a study testing a reminders and incentive intervention.

**Author affiliations**
[1]Wolfson Institute of Population Health, Queen Mary University of London, Asthma UK Centre for Applied Research, London, UK
[2]Biomedical Research Unit at the Royal Brompton and Harefield NHS Foundation Trust and Imperial College London, Asthma UK Centre for Applied Research, London, UK
[3]Centre for Behavioural Medicine, UCL School of Pharmacy - UCL, Asthma UK Centre for Applied Research, London, UK
[4]Unit for Social and Community Psychiatry, Queen Mary University of London, London, UK
[5]Usher Institute - University of Edinburgh, Asthma UK Centre for Applied Research, Edinburgh, UK

**Acknowledgements** We are grateful to Angela Jamalzadeh, Pippa Hall and Sammy Ndlovu-Dawika, members of the direct care team at Royal Brompton Hospital, for invaluable help with the study logistics and liaising with eligible patients.

**Contributors** AS, ADS and CG conceived the work. ADS, LF, RH, SP, AB, AS and CG contributed to the design of the study. ADS wrote the study protocol, and collected the data with LH. ADS analysed the data, and all authors (ADS, LH, RH, LF, AB, CG) contributed to the interpretation of findings. RH helped with the classification and mapping of the emerging themes according to the PAPA framework. ADS wrote the first version of the manuscript. All authors undertook a critical revision of the article and gave their final approval for the manuscript to be published. ADS acts as guarantor.

**Funding** ADS was partly funded by an NIHR Academic Clinical Lectureship and Barts Charity grant reference MGU0419. REAL- Health: REsearch Actionable Learning Health Systems Asthma programme; LH through an NIHR Academic Clinical Fellowship. AB is an NIHR senior investigator and additionally was supported by the NIHR Respiratory Disease. RH was supported by the National Institute for Health Research (NIHR) Collaboration for Leadership in Applied Health Research (CLAHRC) North Thames at Barts NHS Trust. AS acknowledges funding support from the HDRUK BREATHE Hub and the Asthma UK Centre for Applied Research. This work is carried out with the support of BREATHE - The Health Data Research Hub for Respiratory Health [MC_PC_19004] in partnership with Queen Mary University of London. BREATHE is funded through the UK Research and Innovation Industrial Strategy Challenge Fund and delivered through Health Data Research UK. This work uses data provided by the Royal Brompton and Harefield NHS Foundation Trust and Imperial College London.

**Disclaimer** The views expressed are those of the author(s) and not necessarily those of the NHS, NIHR or Department of Health.

**Competing interests** None declared.

**Patient consent for publication** Not applicable.

**Ethics approval** This study received ethical approval from London Bridge Research Ethics Committee (REC reference: 17/LO/1913), IRAS project ID: 207 345.

**Provenance and peer review** Not commissioned; externally peer reviewed.

**Data availability statement** All data relevant to the study are included in the article or uploaded as online supplemental information.

**ORCID iDs**
Anna De Simoni http://orcid.org/0000-0001-6955-0885
Chris Griffiths http://orcid.org/0000-0001-7935-8694

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
