## [Reviewer comments · BMJ Open]

ARTICLE DETAILS

TITLE (PROVISIONAL)	Electronic reminders and rewards to improve adherence to inhaled asthma treatment in adolescents: a non-randomised feasibility study in tertiary care
AUTHORS	De Simoni, Anna; Fleming, Louise; Holliday, Lois; Horne, Robert; Priebe, Stefan; Bush, Andrew; Sheikh, Aziz; Griffiths, Chris

VERSION 1 – REVIEW

REVIEWER	Percefull, Julie Galen College of Nursing
REVIEW RETURNED	12-Jun-2021

GENERAL COMMENTS	Well defined study that was clearly depicted in each element of the manuscript as submitted.
--

REVIEWER	Morton, Robert W. McMaster University
REVIEW RETURNED	25-Jun-2021

GENERAL COMMENTS	Congratulations on completing what sounds like a very difficult study with multiple technical and covid -related obstacles along the way! Overall I like this study, it is well written with the data presented clearly. However, I have reservations as to whether this paper has demonstrated the need for an RCT with the current technology/ approach and would like to see this discussed further. The following are some specific points I'd like to make: 1. Even with a monetary incentive, only 8/12 (66%) participants were retained in the study. Therefore, despite the surveys suggesting the approach is popular, I would take heed of these participants who potentially voted with their feet and didn't take part. This is a very interesting point which should be hi-lighted, that even a cash incentive isn't enough for some teenagers to engage/ take inhalers. I'd like to see this reflected more in the reservations/ discussion.2. It is clear to me the Hallie equipment and software used in this study are not fit for purpose. Previous studies have reported multiple problems with Hallie devices and it is disheartening to read the high number of malfunctions in this study, and lack of ongoing tech support. My major reservation as to whether this work should be developed into a larger RCT would be the reliability of the technology for any larger studies. It is likely that with the current technology further problems will arise, and I take this as one of the conclusions of this study. If reminders or adherence data from devices are not reliable the entire intervention is flawed, as there is no way of knowing whether the
--

	device is faulty or the adherence rates are low. I think this is one of the major findings of this study, to hi-light that we are still not there with reliable technology to monitor adherence via apps. I'd like to see this discussed in more detail if possible, although it doesn't detract from this paper. 3. For me, the data show that whilst this is a potentially useful intervention, it is unpopular in some participants (not recruited/ drop outs, despite cash incentive), and unfeasible in others. Are these the same groups of patients who also have the lowest adherence, and are potentially in the lowest socio-economic groups with higher levels of technology poverty?
--	--

VERSION 1 – AUTHOR RESPONSE

Reviewer: 1

Comments to the Author:

Well defined study that was clearly depicted in each element of the manuscript as submitted.

We thank the reviewer for positive comments.

Reviewer: 2

Congratulations on completing what sounds like a very difficult study with multiple technical and covid -related obstacles along the way!

Overall I like this study, it is well written with the data presented clearly.

We thank the reviewer for positive comments.

However, I have reservations as to whether this paper has demonstrated the need for an RCT with the current technology/ approach and would like to see this discussed further.

The following are some specific points I'd like to make:

1. Even with a monetary incentive, only 8/12 (66%) participants were retained in the study. Therefore, despite the surveys suggesting the approach is popular, I would take heed of these participants who potentially voted with their feet and didn't take part. This is a very interesting point which should be hi-lighted, that even a cash incentive isn't enough for some teenagers to engage/ take inhalers. I'd like to see this reflected more in the reservations/ discussion.

RESPONSE: We acknowledge this is an important point. We have provided some additional clarifications on recruitment, see page 9, lines 11-12:

Of those invited, 10 patients (80% participation response rate) were recruited to the study, 1 parent declined on behalf of the child due to lack of time and having to rush off the clinic, 1 patient did not own a personal mobile phone.

Of the 10 participants recruited, 8 were retained (80% retention).

We have added a sentence to the study limitations within the Discussion, see page 25, lines 16-17:
The retention rate to the study (80%) may suggest that that even a cash incentive is not enough for some teenagers to engage/ take inhalers.

2. It is clear to me the Hallie equipment and software used in this study are not fit for purpose. Previous studies have reported multiple problems with Hallie devices and it is disheartening to read the high number of malfunctions in this study, and lack of ongoing tech support. My major reservation as to whether this work should be developed into a larger RCT would be the reliability of the technology for any larger studies. It is likely that with the current technology

further problems will arise, and I take this as one of the conclusions of this study. If reminders or adherence data from devices are not reliable the entire intervention is flawed, as there is no way of knowing whether the device is faulty or the adherence rates are low. I think this is one of the major findings of this study, to hi-light that we are still not there with reliable technology to monitor adherence via apps. I'd like to see this discussed in more detail if possible, although it doesn't detract from this paper.

RESPONSE: We acknowledge this is an important point. We have discussed this within the Discussion, mentioning also the need of a further study with more reliable EMDs prior to a definitive RCT.

See page 12, lines 18-19:

The lack of reliability of reminders delivery within some EMDs may have had significant effects on the study outcomes, calling for further research with more reliable EMDs prior to a definitive RCT.

3. For me, the data show that whilst this is a potentially useful intervention, it is unpopular in some participants (not recruited/ drop outs, despite cash incentive), and unfeasible in others. Are these the same groups of patients who also have the lowest adherence, and are potentially in the lowest socio-economic groups with higher levels of technology poverty?

RESPONSE: Thank you for this interesting reflection. We are not able to answer this important question with this small study and its numerous limitations. We are planning a pilot trial, with an improved EMD and technical support, which would hopefully shed more light on the role of electronic reminders and rewards to improve adherence to inhaled asthma treatment in adolescents.